# Informative and adaptive distances and summary statistics in sequential approximate Bayesian computation

**Yannik Schälte** [1,2,3], **Jan Hasenauer** [1,2,3]*

**1** Faculty of Mathematics and Natural Sciences, Rheinische Friedrich-Wilhelms-Universität Bonn, Bonn, Germany, **2** Institute of Computational Biology, Helmholtz Zentrum München, Neuherberg, Germany, **3** Center for Mathematics, Technische Universität München, Garching, Germany

* jan.hasenauer@uni-bonn.de

**Data Availability Statement:** The code underlying the application study is on GitHub (https://github.com/yannikschaelte/study_abc_slad), a snapshot of code and data on Zenodo (http://doi.org/10.5281/zenodo.5522919). All methods developed in

## Abstract

Calibrating model parameters on heterogeneous data can be challenging and inefficient. This holds especially for likelihood-free methods such as approximate Bayesian computation (ABC), which rely on the comparison of relevant features in simulated and observed data and are popular for otherwise intractable problems. To address this problem, methods have been developed to scale-normalize data, and to derive informative low-dimensional summary statistics using inverse regression models of parameters on data. However, while approaches only correcting for scale can be inefficient on partly uninformative data, the use of summary statistics can lead to information loss and relies on the accuracy of employed methods. In this work, we first show that the combination of adaptive scale normalization with regression-based summary statistics is advantageous on heterogeneous parameter scales. Second, we present an approach employing regression models not to transform data, but to inform sensitivity weights quantifying data informativeness. Third, we discuss problems for regression models under non-identifiability, and present a solution using target augmentation. We demonstrate improved accuracy and efficiency of the presented approach on various problems, in particular robustness and wide applicability of the sensitivity weights. Our findings demonstrate the potential of the adaptive approach. The developed algorithms have been made available in the open-source Python toolbox pyABC.

## 1 Introduction

Mechanistic models are important tools in systems biology and many other research areas to describe and study real-world systems, allowing researchers to understand underlying mechanisms [1, 2]. Commonly, they are subject to parameters that need to be estimated by comparison of model outputs to observed data [3]. The Bayesian framework allows doing so by combining the likelihood of data and prior information on parameters. However, for complex stochastic models, e.g. used in systems biology to describe multi-cellular systems, evaluating the likelihood is often computationally infeasible [4, 5]. Therefore, likelihood-free methods

this contribution have been implemented, tested and documented in the open-source Python package pyABC (https://github.com/icb-dcm/pyabc).

**Funding:** This work was supported by the German Federal Ministry of Education and Research (BMBF) (FitMultiCell/031L0159 and EMUNE/031L0293) and the German Research Foundation (DFG) under Germany's Excellence Strategy (EXC 2047 390873048 and EXC 2151 390685813) and a Schlegel Professorship for JH. YS acknowledges support by the Joachim Herz Foundation. The funders had no role in study design, data collection and analysis, decision to publish, or preparation of the manuscript.

**Competing interests:** The authors have declared that no competing interests exist.

such as approximate Bayesian computation (ABC) have been developed [6, 7]. Put briefly, in ABC likelihood evaluation is circumvented by simulating data, and accepting these depending on their proximity to observed data, according to a distance measure and an acceptance threshold. In this way, it generates samples from an approximation to the posterior distribution. ABC is frequently combined with a sequential Monte Carlo scheme (ABC-SMC) [8, 9], which allows gradually reducing the acceptance threshold while maintaining high acceptance rates.

ABC relies on the comparison of relevant features in simulated and observed data. [10] demonstrates superior performance of distances that adaptively weight model outputs to normalize contributions on different scales, exploiting the structure of ABC-SMC algorithms. In [11], we extend this approach to outlier-corrupted data. However, an implicit assumption of scale normalization is that all model outputs are similarly informative of the parameters. It can worsen performance, e.g. when inflating the impact of data points underlying only background noise. Therefore, it would be preferable to either only consider informative statistics, or to account for informativeness in the weighting scheme.

Especially for noise-corrupted high-dimensional data, often lower-dimensional summary statistics are employed [12]. Various methods to construct such statistics have been developed, e.g. via subset selection or auxiliary likelihoods [13, 14]. In a popular class of approaches, inverse regression models of parameters on simulated data have been used as statistics [15–17]. Here, by "inverse" we mean that the summary statistics map from (functions of) simulated data back to (functions of) the parameters, i.e. in the inverse direction to the forward mechanistic model. Such regression models can be heuristically motivated as summarizing the information in the data in a single value per parameter. In addition, [15] argue that the resulting summary statistics effectively approximate posterior means, which conserves the true posterior mean in the ABC analysis.

To evaluate proximity of regression-based statistics, e.g. Euclidean distances have been used, or weighted Euclidean distances using weights based on calibration samples [15]. However, here essentially the same problems apply that motivated the use of adaptive weighting [10], shifted from the level of data to the level of parameters, or regression approximations thereof. In fact, the approach by [10] is particularly applicable to regression-based statistics, as all outputs are informative. A further problem with regression-based statistics is that a unique inverse mapping from data to parameters may not always exist even in the noise-free limit, resulting e.g. in a multi-modal posterior distribution, or plateaus of combinations of parameter values achieving the same posterior value. This is in particular the case when the observable model outputs and data are not rich enough to allow to structurally identify the parameters globally [18].

In this work, we present two approaches combining the concepts of adaptive distances and regression models. First, we integrate summary statistics learning in an ABC-SMC framework with scale-normalizing adaptive distances. Second, the focus of this work, we employ regression models not to transform data, but in order to inform additional sensitivity weights that account for informativeness. Moreover, we discuss the problem of non-identifiability of the inverse mapping, and present a solution using augmented regression targets. On a dedicated test problem exhibiting multiple problematic features such as partly uninformative data, heterogeneous data and parameter scales, and non-identifiability, we demonstrate how both scale-normalizing distances [10], and regression-based summary statistics [15] fail to approximate the true posterior distribution. Then, we demonstrate substantially improved performance of the newly introduced approaches. We evaluate the proposed methods on further test problems, including a systems biology application example and outlier-corrupted data,

demonstrating in particular robustness as well as wide applicability of the sensitivity-weighted distance.

## 2 Methods

### 2.1 Background

In this section, we give required background knowledge on the underlying methodology.

**2.1.1 Approximate Bayesian computation.** In Bayesian inference, the likelihood $\pi(y_{\text{obs}}|\theta)$ of observing data $y_{\text{obs}} \in \mathbb{R}^{n_y}$ under model parameters $\theta \in \mathbb{R}^{n_\theta}$ is combined with prior information $\pi(\theta)$, giving the posterior $\pi(\theta|y_{\text{obs}}) \cdot \pi(y_{\text{obs}}|\theta) \cdot \pi(\theta)$. Here and throughout this manuscript, $n_{\text{var}}$ denotes the dimension of a given variable var. We assume that while numerical evaluation of $\pi(y_{\text{obs}}|\theta)$ is infeasible, the model is generative, i.e. allows to simulate data $y \sim \pi(y|\theta)$. The core principle of ABC consists of three steps [6]:

1. Sample parameters $\theta \sim \pi(\theta)$.

2. Simulate data $y \sim \pi(y|\theta)$.

3. Accept $(\theta, y)$ if $d(y, y_{\text{obs}}) \leq \varepsilon$.

Here, the distance $d : \mathbb{R}^{n_y} \times \mathbb{R}^{n_y} \to \mathbb{R}_{\geq 0}$ compares simulated and observed data, and $\varepsilon \geq 0$ an acceptance threshold. This is repeated until sufficiently many, say $N$, particles have been accepted.

For high-dimensional data, the comparison is often in terms of summary statistics $s : \mathbb{R}^{n_y} \to \mathbb{R}^{n_s}$, as $d(s(y), s(y_{\text{obs}})) \leq \varepsilon$, with $d : \mathbb{R}^{n_s} \times \mathbb{R}^{n_s} \to \mathbb{R}_{\geq 0}$ and typically $n_s \ll n_y$. Denoting $\pi(s|\theta) \propto \int I[s(y) = s]\pi(y|\theta)\, dy$ the intractable summary statistics likelihood with $I$ the indicator function, and $s_{\text{obs}} = s(y_{\text{obs}})$, the population of accepted particles then constitutes a sample from the approximate posterior distribution

$$\pi_{\text{ABC}}(\theta|s_{\text{obs}}) \propto \int I[d(s, s_{\text{obs}}) \leq \varepsilon]\pi(s|\theta)ds \cdot \pi(\theta),$$

where $\pi_{\text{ABC}}(s_{\text{obs}}|\theta) \propto \int I[d(s, s_{\text{obs}}) \leq \varepsilon]\pi(s|\theta)ds$ can be interpreted as an approximation to the likelihood.

For $\varepsilon \to 0$, it holds under mild assumptions that $\pi_{\text{ABC}}(\theta|s(y_{\text{obs}})) \to \pi(\theta|s(y_{\text{obs}})) \cdot \pi(s(y_{\text{obs}})|\theta)\pi(\theta)$ in an appropriate sense [19]. Compared to likelihood-based sampling, ABC introduces two approximation errors [20, Chapter 1]. First, it accepts not only particles with $y = y_{\text{obs}}$, which occur for continuous models with probability zero, but also proximate ones according to $d$. Second, only for sufficient statistics, $\pi(\theta|s_{\text{obs}}) \equiv \pi(\theta|y_{\text{obs}})$, is the original posterior recovered in the approximate limit $\varepsilon \to 0$. In practice, $s$ is however usually insufficient, only capturing essential information about $y$ in a low-dimensional representation.

In this work, we will denote by $y$ either the raw data, or assume it to already incorporate a mapping to (manually crafted) summary statistics. We will assess $y$ in ABC directly via the use of weighted distance metrics, or on top of $y$ automatically derive regression-based summary statistics $s$.

**2.1.2 Sequential importance sampling.** As the above vanilla ABC algorithm, also called ABC-Rejection, exhibits a trade-off between decreasing the acceptance threshold $\varepsilon$ to improve the posterior approximation, and maintaining high acceptance rates, it is frequently combined with a sequential Monte Carlo (SMC) importance sampling scheme [8, 9]. In ABC-SMC, a series of particle populations $P_t = \{(\theta_i^t, y_i^t, w_i^t)\}_{i \leq N}$, $t = 1, \ldots, n_t$, are generated, with acceptance thresholds $\varepsilon_1 > \ldots > \varepsilon_{n_t}$, targeting successively better posterior approximations. Particles for

generation $t$ are sampled from a proposal distribution $g_t(\theta) \gg \pi(\theta)$ based on the previous generation's accepted particles $P_{t-1}$, e.g. via a kernel density estimate, only initially $g_1(\theta) = \pi(\theta)$. The importance weights $w_i^t$ are the corresponding non-normalized Radon-Nikodym derivatives, $w_t(\theta) = \pi(\theta)/g_t(\theta)$.

**Algorithm 1** A basic ABC-SMC algorithm.

```
initialize ε₁ via calibration samples, let g₁(θ) = π(θ)
for t = 1, ..., nₜ do
  while less than N acceptances do
    sample parameter θ ∼ gₜ(θ)
    simulate data y ∼ π(y|θ)
    accept θ if d(y, y_obs) ≤ εₜ
  end while
  compute weights wᵢᵗ = π(θᵢᵗ)/gₜ(θᵢᵗ), for accepted parameters {θᵢᵗ}ᵢ≤ₙ
  normalize weights Wᵢᵗ = wᵢᵗ/∑ⱼwⱼᵗ
  update gₜ₊₁ and εₜ₊₁ based on particles from generation t
  end for
output: weighted samples {(θᵢⁿᵗ, Wᵢⁿᵗ)}ᵢ≤ₙ
```

The underlying ABC-SMC algorithm (Algorithm 1) used throughout this work is based on [21], using an adaptive threshold scheme based on the median of distances in the previous generation [22] and multivariate normal proposal distributions with adaptive covariance matrix [23], see [24, 25] for details. There exist various ABC-SMC sampler variants [20], e.g. in some cases different threshold schemes [26] or proposal distributions [23] may be beneficial. The distances and summary statistics presented in this work are mostly independent of the sequential sampler specifics.

**2.1.3 Adaptive distances.** A common choice of distance $d$ is a weighted Minkowski distance

$$d(y, y_{\text{obs}}) = \| r \cdot (y - y_{\text{obs}}) \|_p = \left( \sum_{i_y=1}^{n_y} |r_{i_y} \cdot (y_{i_y} - y_{\text{obs},i_y})|^p \right)^{1/p}, \tag{1}$$

with $p \geq 1$ and weights $r_{i_y}$. Frequently, simply unit weights $r = 1$ are used (e.g. [15–17, 21]). However, model outputs can be and vary on different scales, in which case highly variable ones dominate the acceptance decision. This can be corrected for by the choice of weights $r_{i_y}$ in (1), commonly as inversely proportional to measures of variability,

$$r_{i_y} = 1/\sigma_{i_y}, \tag{2}$$

with $\sigma_{i_y}$ e.g. given via the median absolute deviation (MAD) from the sample median [27]. To define weights, calibration samples can be used (e.g. [7, 15]). However, [10] demonstrates that in an ABC-SMC framework, the relative variability of model outputs in later generations can differ considerably from pre-calibration. Thus, they propose an iteratively updated distance $d_t$, defining weights for generation $t$ based on all samples generated in generation $t-1$.

In [11], we demonstrate the L2 norm used in (1) in [10] to be sensitive to data outliers, and show an L1 norm to be more robust on both outlier-corrupted and outlier-free data. To further reduce the impact of outliers, we complement MAD, as a measure of sample variability, by the median absolute deviation to the observed value, as a measure of deviation, giving a normalization term PCMAD ("perhaps use the combined median absolute deviation (CMAD) to the population median (MAD) and to the observation (MADO), or only use MAD"; see [11] for details).

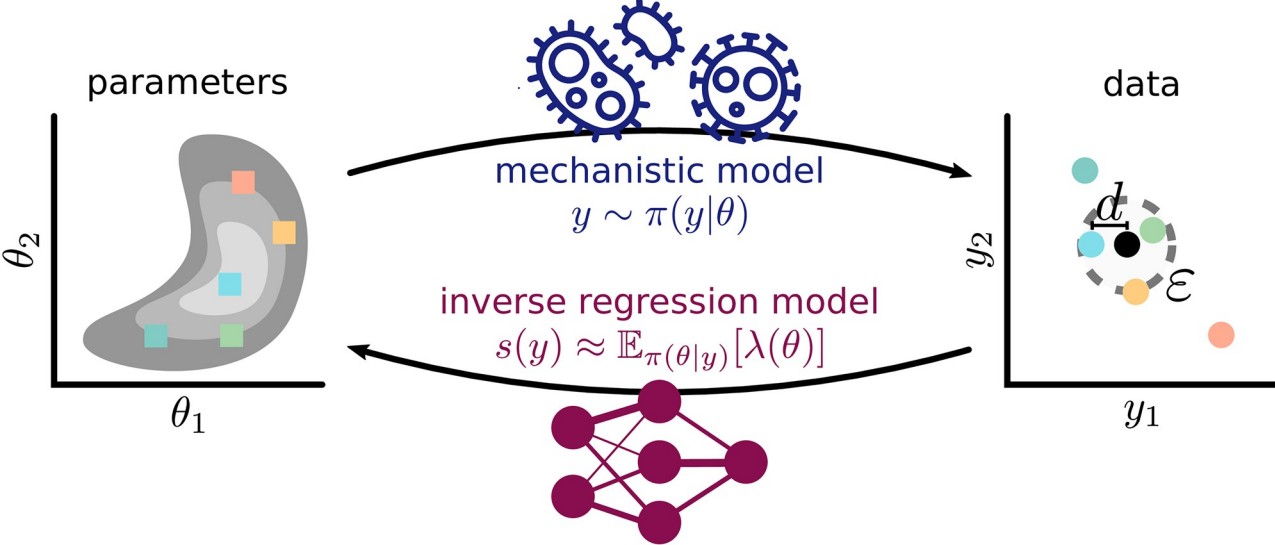

**Fig 1. Concept visualization.** While, given parameters $\theta$, the mechanistic model $\pi(y|\theta)$ simulates data $y$, which are then compared to the observed data (black) via some distance $d$ and threshold $\varepsilon$, we employ regression models to learn an inverse mapping $s$, to either construct summary statistics, or define sensitivity weights for distance calculation.

**2.1.4 Regression-based summary statistics.** The comparison of simulations and data in ABC is often in terms of low-dimensional, informative summary statistics. The "semi-automatic ABC" approach by [15] uses the outputs of a regression model $s : \mathbb{R}^{n_y} \to \mathbb{R}^{n_\theta}$, predicting parameters from simulated data (Fig 1):

1. In an ABC pilot run, determine a high-density posterior region $H$.

2. Generate a population $P = \{(\theta_i, y_i)\}_{i \le \tilde{N}} \sim \pi(y|\theta)I[\theta \in H]$, for some $\tilde{N} \in \mathbb{N}$.

3. Train a regressor model $s : \mathbb{R}^{n_y} \to \mathbb{R}^{n_\theta}, y \mapsto \theta$, on $P$.

4. Run the actual ABC analysis using $s$ as summary statistics.

In step 4, the distance operates on $s(y)$. Step 1 aims to find a good training region, and can be skipped for informative priors [15]. In [17], $H = [0.5\tilde{\theta}, 2\tilde{\theta}]$ is used, around a literature value $\tilde{\theta}$, based on manual experimentation, which is in practice only applicable if reliable references exist. In [16], step 1 is omitted, using the prior directly, in one case constrained to an identifiable region. In step 3, [15] employ a linear regression (LR) model on potentially augmented data. [16] and [17] respectively use neural networks (NN) and Gaussian processes (GP) instead, aiming at a more accurate description of non-linear relationships, and further process automation. The sufficient performance of LR observed in [15] may be due to the substantial time spent in the pilot run, identifying a high-density region where a linear approximation suffices, while e.g. [16] observe a clearly better posterior approximation with NN, and [17] better model predictions with GP.

A theoretical justification of regression-based summary statistics is that the regression model serves as an approximation to the posterior mean, $s(y) \approx \mathbb{E}_{\pi(\theta|y)}[\theta]$, using which as statistic ensures that the ABC posterior approximation recovers the actual posterior mean as $\varepsilon \to 0$, see [15, 16], or S1 File, Theorem 1.

## 2.2 Adaptive and informative regression-based distances and summary statistics

In this section, we describe the novel methods introduced in this work.

**2.2.1 Integrating summary statistics learning and adaptive distances.** In previous studies, the regression approach from Section 2.1.4 was used together with uniform, or on a previous run pre-calibrated, distance weights [15–17]. However, to the regression model outputs, approximating underlying parameters, the same problems apply that motivated the adaptive approach in [10]: Parameters varying on larger scales dominate the analysis without scale adjustment, with potentially changing levels of variability over ABC-SMC generations.

We propose to combine the regression-based summary statistics from Section 2.1.4 with the weight adaptation from Section 2.1.3. The regression model can be pre-trained as previously done. Here, we however suggest to increase efficiency and automation by integrating the training into the actual ABC-SMC run (Algorithm 2). We begin by using an adaptively scale-normalized distance on the full model outputs. Then, in a generation $t_{\text{train}} \geq 1$, the regression model $s : \mathbb{R}^{n_y} \to \mathbb{R}^{n_\theta}$ is trained on all particles $\{(\theta_i^{t_{\text{train}}-1}, y_i^{t_{\text{train}}-1})\}_{i \leq \tilde{N}}, \tilde{N} \geq N$, generated in the previous generation. From $t \geq t_{\text{train}}$ onward, the regression model outputs $s(y)$ are used as summary statistics, also here using a scale-normalized distance with iteratively adjusted weights. Like for adaptive distance weight calculation, the training samples also include rejected ones. First, this increases the training sample size, and second, it gives a representative sample from the joint distribution of data and parameters, focusing on a high-density region, but not confined to $y \approx y_{\text{obs}}$.

The delay of regression model training until after a few generations serves to focus on a high-density region, similar to [15], such that simpler regression models provide a sufficient description. While [15] update the prior to a typical range of values observed in the pilot run, we consider the prior as part of the problem formulation, and thus do not update it. In generations $t \geq t_{\text{train}}$ the proposal distributions $g_t$ will usually anyway mostly suggest values within the training domain range.

**Algorithm 2** ABC-SMC algorithm with regression-based summary statistics or sensitivity-weighted distances.

```
initialize ε₁, σ¹ᵢᵧ via calibration samples, let g₁(θ) = π(θ)
for t = 1, ..., nₜ do
  while less than N acceptances do
    sample parameter θ ~ gₜ(θ)
    simulate data y ~ π(y|θ)
    if t < t_train then
      accept θ if dₜ(y, y_obs) ≤ εₜ, where dₜ uses scale weights rᵗᵢᵧ = 1/σᵗᵢᵧ
    else if s is used as summary statistics then
      accept θ if dₜ(s(y), s(y_obs)) ≤ εₜ, where dₜ uses scale weights
rᵗᵢₛ = 1/σᵗᵢₛ
    else if using s only to define sensitivity weights qᵢᵧ then
      accept if dₜ(y, y_obs) ≤ εₜ, where dₜ uses scale and sensitivity
weights rᵗᵢᵧ = qᵗᵢᵧ/σᵗᵢᵧ
    end if
  end while
  compute importance weights wᵗᵢ = π(θᵗᵢ)/gₜ(θᵗᵢ), for accepted parameters {θᵗᵢ}ᵢ≤N
normalize importance weights Wᵗᵢ = wᵗᵢ/∑ⱼ wᵗⱼ
  if t + 1 == t_train then
    train regression model s on all particles from generation t
    if using s to weight model outputs then
      define sensitivity weights q₁,...,qₙᵧ via s
```

```
    end if
  end if
    update g_{t+1} and ε_{t+1} based on particles from generation t
    update inverse scale weights σ_{i_y}^{t+1} or σ_{i_s}^{t+1} based on all particles from
generation t
end for
output: weighted samples {(θ_i^{n_t}, W_i^{n_t})}_{i≤N}
```

**2.2.2 Regression-based sensitivity weights.** The adaptive scale-normalized distance approach from [10, 11] is, operating on the full data without summary statistics, not ideal if data points are not similarly informative. The regression approach from Section 2.1.4 is one solution to focus on informative statistics. However, it performs a complex transformation of the model outputs, which can hinder interpretation, and perform badly if the regression model is inaccurate. In this section, we present an alternative approach, using the regression model to inform additional weights on the full data, instead of constructing summary statistics. The idea is to weight a data point by how informative it is of underlying parameters. We quantify informativeness via the sensitivity of how much the posterior expectation of parameters, or transformations thereof, given observed data $y_{\text{obs}}$, would vary under data perturbations. As in Section 2.2.1, we use a regression model to describe the inverse mapping from data to parameters.

Specifically, before a generation $t_{\text{train}}$, we train a regression model $s : \mathbb{R}^{n_y} \to \mathbb{R}^{n_\theta}$ on samples from the previous generation. As regression model inputs, we use normalized simulations $y/\sigma_{t_{\text{train}}}$, with $\sigma_{t_{\text{train}}}$ the measure of scale used for distance scale normalization, e.g. MAD. Further, we z-score normalize regression model targets $\theta$, in order to render the model scale-independent. Then, we calculate the sensitivity matrix

$$S = \nabla_y s(y_{\text{obs}}) \in \mathbb{R}^{n_y \times n_\theta} \tag{3}$$

at the observed data. To robustly approximate derivatives, we employ central finite differences with automatic step size control [28]. We define the *sensitivity weight* of model output $i_y$ as

$$q_{i_y} = \sum_{i_\theta=1}^{n_\theta} \frac{|S_{i_y,i_\theta}|}{\sum_{j_y=1}^{n_y} |S_{j_y,i_\theta}|}, \tag{4}$$

i.e. as the sum over the absolute sensitivities of all parameters with respect to the model output, normalized per parameter to level their impact. The normalization can be omitted, but yields more conservative weights, accounting for the fact that the regression model may be inaccurate, by more evenly distributed weights when all sensitivities with respect to some parameters are small. Here, $S_{i_y,i_\theta}$ denotes the sensitivity of the $i_\theta$th output with respect to the $i_y$th input.

The final weight used in the distance (1) is then given as the product of scale weight (2) and sensitivity weight (4),

$$r_{i_y} = q_{i_y}/\sigma_{i_y}, \tag{5}$$

with here $\sigma_{i_y}$ e.g. again given via MAD, or, also taking bias into account, PCMAD. This separate treatment of scale and sensitivity weights allows to e.g. include the error correction from [11] in the scale correction, but not in the normalized data used for regression model training, which would lead to inversely re-scaled sensitivities. Thus, we can simultaneously account for informativeness and outliers. As long as $r_{i_y} \neq 0$ for all weights, the original posterior $\pi(\theta|y_{\text{obs}})$ can be conceptually recovered for $\varepsilon \to 0$ [10, 19], i.e. no information is lost, unlike for insufficient summary statistics, while practical convergence is clearly weight-dependent.

**2.2.3 Optimal summary statistics to recover distribution features.**   A problem with inverse regression models of parameters on data is that such a mapping may not exist. For example, consider a quadratic model $y \sim \mathcal{N}(\theta^2, 0.1^2)$, with prior $\theta \sim U[-1, 1]$, and observed data $y_{\text{obs}} = 0.7$. Given the data, the underlying parameter cannot be uniquely identified. As an inverse mapping $y \mapsto \theta$ does not exist globally, a regression model $s: y \mapsto \theta$ cannot extract a meaningful relationship. Indeed, the problem is symmetric in $\theta$, such that the posterior mean is $\mathbb{E}_{\pi(\theta|y)}[\theta] = 0$, using which as summary statistic as in [12, 15] would clearly recover the true posterior mean. However, it would fail to describe the posterior shape at all. More generally, an informative inverse mapping from data back to parameters can only exist when the forward model $\theta \mapsto \pi(\cdot|\theta)$ is injective, i.e. the model is structurally identifiable in the limit of infinite data [29, 30]. We would like to remark that this lack of identifiability of true parameters given data also stands in the way of theoretical asymptotic results obtained for ABC methods regarding convergence of point estimators in the large-data limit in [31] (Condition 4.(ii)) and limiting shape of posteriors and posterior means in [32] (Assumption 3.(ii)).

In general, non-identifiability can be tackled in various ways, e.g. by fixing some parameters to constant values, reparameterization, or by generating additional data to break the symmetry [18]. However, this requires an in-depth and iterative model analysis. Even for deterministic ordinary differential equation based models, this is already a challenging task [33], while ABC is commonly applied to highly complex stochastic models. Further, one ends up with an altered inference problem, which renders the analysis of prediction uncertainties more involved [33].

Here we propose an approach that is able to work with the original, unaltered, problem formulation, including potential non-identifiabilities. We solve the problem by considering transformations $\lambda(\theta)$ of the parameters, e.g. higher-order moments $s: y \mapsto \lambda(\theta) = (\theta^1, \dots, \theta^k)$, which may be better described as functions of the data, or identifiable in the first place. In the above example, it suffices to consider $\theta^2$, giving a linear mapping $y \sim \theta^2$ and breaking the symmetry. While the use of parameter transformations as regression model targets is heuristically reasonable, their use can be theoretically further justified: Employing as summary statistics posterior expectations of transformations of the parameters,

$$s(y) = \mathbb{E}_{\pi(\theta|y)}[\lambda(\theta)],$$

allows under mild assumptions to recover the corresponding posterior expectations for $\varepsilon \to 0$,

$$\lim_{\varepsilon \to 0} \mathbb{E}_{\pi_{\text{ABC},\varepsilon}}[\lambda(\Theta)|s(y_{\text{obs}})] = \mathbb{E}[\lambda(\Theta)|Y = y_{\text{obs}}],$$

see Theorem 1 in S1 File for details.

Obviously, conditional posterior expectations are hardly available in practice. However, we may interpret the above regression-based summary statistics as approximations, aiming at a sufficiently accurate description of the underlying expectations by the regression model. Thus, we propose to use $\lambda(\theta)$ as targets, both for summary statistics (Section 2.2.1), and sensitivity weights (Section 2.2.2).

We would like to remark that here we augment the regression targets $\theta$, while e.g. [12, 15] instead augment the regressors $y$, e.g. by higher-order moments $y \mapsto (y^1, \dots, y^4)$. Their approach allows to employ simple regression models, such as linear ones, on an increased feature space, which can be guided by model analysis and prior experimentation. However, it can conceptually not solve the problem tackled by our approach, that some parameters cannot be described as functions of the data (or transformations thereof). Of course, it would also be possible to augment regressors and regression targets simultaneously.

## 2.3 Implementation

We implemented all presented methods in the open-source Python package pyABC (https://github.com/icb-dcm/pyabc) [25], interfacing particularly scikit-learn regression models [34]. The code underlying the application study is on GitHub (https://github.com/yannikschaelte/study_abc_slad), a snapshot of code and data on Zenodo (http://doi.org/10.5281/zenodo.5522919).

# 3 Results

We evaluated the performance of the proposed methods on various test problems.

## 3.1 Distances and summary statistics

As distance to compare model outputs or summary statistics, we considered, given its robust performance in [11], an L1 norm, with adaptive MAD weights when employing scale-normalization (denoted "Ada.+MAD"). Acceptance in generation $t$ was only based on $d_t$, but not previous acceptance criteria, for ease of implementation and as for an L1 norm no substantial differences were observed in [11].

As regression models, we considered LR and NN. We trained the regression model after 40% of the simulation budget. For comparison, we also considered training the regression model before the initial generation, $t_{\text{train}} = 1$, based on samples from the prior ("Init"). NN models were considered with a single hidden layer of dimension $[(n_y + n_\theta)/2]$, with ReLU activation function, using ADAM stochastic gradient descent for optimization, and early stopping to avoid overfitting, with a 10% validation set. Both regression models were computationally efficient compared to the full ABC-SMC analyses, with run-times on the order of milliseconds (LR) or few seconds (NN).

When employing parameter augmentation (Section 2.2.3), we used the first four moments, $\lambda(\theta) = (\theta^1, \ldots, \theta^4)$ ("P4"). We considered both regression to define summary statistics ("Stat", Section 2.2.1) and sensitivity weights ("Sensi", Section 2.2.2). We used a single regression model to learn the full vector-valued mapping $y \mapsto \lambda(\theta)$, allowing to leverage synergies, as opposed to learning a separate model per regression target.

For example, L1+Ada.+MAD+StatNN denotes an analysis using consistently an adaptive distance with MAD normalized weights, and using a neural network to construct summary statistics after 40% of the total simulation budget, with regression targets $\lambda(\theta) = \theta$. L1+Ada.+MAD+SensiLR+P4 uses an adaptive distance with scale-normalizing weights via MAD, and a linear model to define further sensitivity weights, with regression targets $\lambda(\theta) = (\theta^1, \ldots, \theta^4)$, and L1+StatLR uses a linear model for summary statistics construction, but uses uniform distance weights.

## 3.2 Performance on dedicated demonstration problem

To illustrate the different problems addressed in this work, we constructed a demonstration problem with four parameters and five types of data, constituting a joint data vector $y = (y_1, \ldots, y_5) \in \mathbb{R}^{17}$:

- $y_1 \sim \mathcal{N}(\theta_1, 0.1^2)$ is informative of $\theta_1$, with a relatively wide prior $\theta_1 \sim U[-7, 7]$,

- $y_2 \sim \mathcal{N}(\theta_2, 100^2)$ is informative of $\theta_2$, with prior $\theta_2 \sim U[-700, 700]$,

- $y_3 \sim \mathcal{N}(\theta_3, 4 \cdot 100^2)^{\otimes 4} \in \mathbb{R}^4$ is informative of $\theta_3$, with prior $\theta_3 \sim U[-700, 700]$,

- $y_4 \sim \mathcal{N}(\theta_4^2, 0.1^2)$ is informative of $\theta_4$, with prior $\theta_4 \sim U[-1, 1]$, but quadratic in the parameter,

- $y_5 \sim \mathcal{N}(0, 10)^{\otimes 10} \in \mathbb{R}^{10}$ is uninformative.

The model dynamics are purposely simple, such that inverse mappings can be captured easily. The problem exhibits the following potentially problematic features:

- A substantial part of the data, $y_5$, is uninformative, such that approaches ignoring data informativeness may converge slower.

- Both data and parameters are on different scales, such that approaches comparing data, or, via regression-based summary statistics, parameters, without normalization focus on large-scale variables. Further, e.g. the prior of $\theta_1$ is relatively wide, preventing pre-calibration.

- $y_4$ is quadratic in $\theta_4$, such that first-order regression models cannot capture a meaningful relationship.

- While $y_2, y_3$ are such that the posteriors of $\theta_2, \theta_3$ are identical, in solely scale-normalized approaches, the impact of $y_4$ on the distance value is roughly four times as high as that of $y_3$, resulting in uneven convergence.

We studied the demonstration problem with synthetic data $y_{\text{obs},1}, y_{\text{obs},2}, y_{\text{obs},3}, y_{\text{obs},5} \equiv 0$, $y_{\text{obs},4} = 0.7$, using a population size of $N = 4e3$ with a total budget of $1e6$ simulations per run. Marginal posterior approximations obtained using selected distances and summary statistics are shown in Fig 2.

**Solely scale-normalized distances without informativeness assessment converge slowly.** The scale-normalized adaptive distance L1+Ada.+MAD correctly captured all posterior modes and shapes, in particular the bi-modality of $\theta_4$, however with large variances, because the uninformative model outputs $y_5$ were considered on the same scale as the informative ones (Fig 2 bottom). Further, while the true posteriors of $\theta_2$ and $\theta_3$ coincide, L1+Ada.

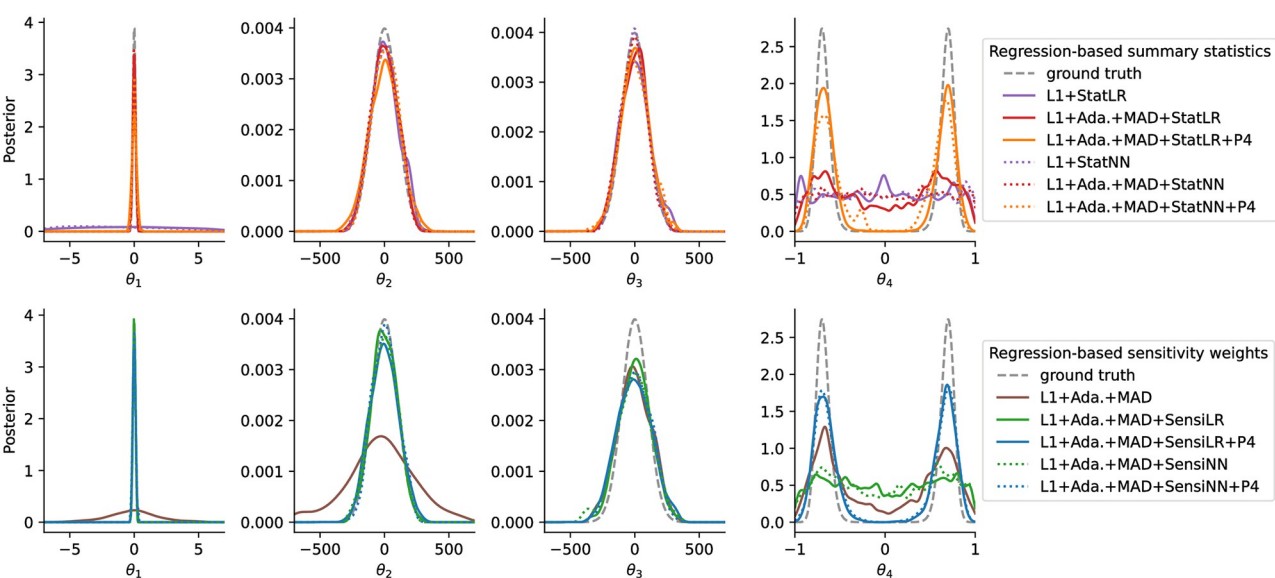

**Fig 2.** ABC marginal posterior approximations obtained using regression-based summary statistics (top, "Stat") or sensitivity weights (bottom, "Sensi") on the demonstration problem, using an underlying L1 norm, uniformly weighted, or MAD scale-normalized distance weights ("Ada.+MAD"), using a linear ("LR") or a neural network ("NN") regression model, and in some cases augmented regression targets $\theta^1, \ldots, \theta^4$ ("P4").

+MAD assigned a substantially wider variance to $\theta_2$, as only a single model output, $y_2$, is informative of it, while four are of $\theta_3$, all on the same normalized scale.

**Non-scale-normalized distances converge unevenly.** The analyses L1+Stat{LR/NN} without scale normalization described the posteriors of $\theta_2$ and $\theta_3$ accurately, which are on the same scale, however yielded substantially wider variances for $\theta_1$ (Fig 2 top), because $\theta_1$, used as regression target, varies on a smaller scale. In contrast, all analyses employing scale normalization described $\theta_1$, $\theta_2$, and $\theta_3$ roughly or almost similarly well, with the exception of L1+Ada.+MAD, as outlined above.

**Regression models not accounting for non-identifiability cannot capture posterior.** All analyses employing regression models but using the non-augmented regression targets $\lambda(\theta) = \theta$ failed to describe the bi-modal distribution of $\theta_4$, because a global mapping $y_4 \mapsto \theta_4$ does not exist. In comparison, analyses considering higher-order regression targets ("P4") captured the bi-modality, as for this problem a linear mapping $\theta_4^2 \sim y_4$ exists, or a quadratic one $\theta_4^4 \sim y_4^2$.

**Novel approaches fit all parameters well.** The analyses L1+Ada.+MAD+{Stat{LR/NN}/ Sensi{LR/NN}}+P4 combining all methods introduced in this work, i.e. scale normalization, informativeness assessment via regression-based summary statistics or sensitivity weights, and regression target augmentation, provided the overall best description of all posterior marginals, with roughly homogeneously small variances. Advantages of NN over LR were not observed.

Estimates for $\theta_3$ were with L1+Ada.+MAD+Sensi{LR/NN} consistently slightly worse than with L1+Ada.+MAD+Stat{LR/NN}. This can be explained by the latter approaches employing a one-dimensional interpolation of $y_3 \in \mathbb{R}^4$, and thus e.g. an approximation of the sufficient statistic $\frac{1}{4}\sum_{i=1}^{4} y_{3,i}$. Meanwhile, approaches that do not transform but only weight, are more subject to random noise. This illustrates that when low-dimensional sufficient statistics exist and are accurately captured, employing explicit dimension reduction can be superior to mere re-weighting.

**Sensitivity weights permit further insights.** In Fig 3, normalized absolute sensitivities (4) of parameters with respect to model outputs are visualized. Overall, both regression models captured the relationship of model outputs and parameters well, and assigned large, albeit not completely homogeneous, sensitivity weights to $y_1, \ldots, y_4$, and lower ones to $y_5$, with roughly $q_1 \approx q_2 \approx \sum_{i=1}^{4} q_{3,i}$. The description provided by NN was overall slightly better than LR, assigning lower weights to $y_5$, and capturing the non-linear mappings $\theta_1^2 \sim y_1$ and $\theta_1^4 \sim y_1$ better. As seen above, LR nevertheless sufficed to yield good posterior approximations. Sensitivities of $\theta_4^1$ and $\theta_4^3$ were, as expected, comparably small with respect to all variables.

The weight assigned to $y_4$ was roughly half the ones assigned to $y_1$, $y_2$ and $y_3$, because $\theta_4^1$ and $\theta_4^3$ could not be accurately described. Correspondingly, the variance of $\theta_4$ was slightly wider under sensitivity-weighted analyses, compared to using summary statistics (Fig 2). This could be improved by not employing parameter-wise normalization in (4), which however makes the analysis less robust to regression model misspecification, or by an alternative normalization.

An analysis such as performed here may generally allow evaluating regression model plausibility, and to obtain insights into parameter-data relationships, e.g. eliciting uninformative data.

## 3.3 Performance on general test problems

To evaluate robustness and general performance of the proposed methods, we next considered six test problems T1–6, not tailored to the challenges discussed in Section 3.2. Core model

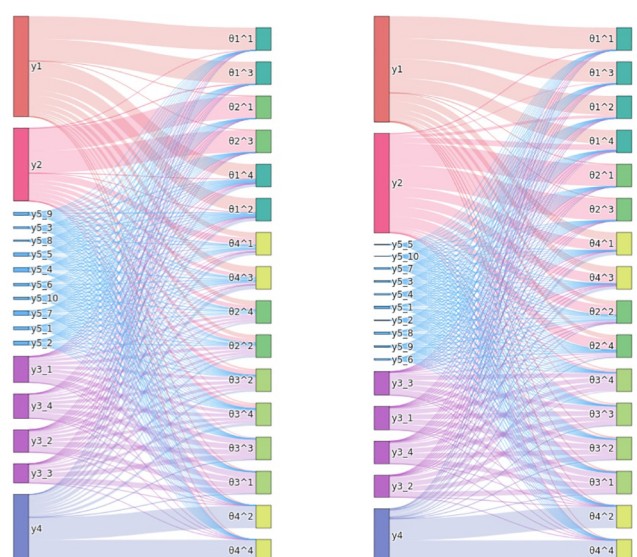

**Fig 3.** Exemplary normalized absolute data-parameter sensitivities for the demonstration problem, using LR (left) and NN (right), for all regression targets $\theta^1, \ldots, \theta^4$ with $\theta = (\theta_1, \ldots, \theta_4)$ (respectively on the right), with respect to all data coordinates $y$ (respectively on the left). The absolute sensitivity matrix $|S|$ (3) was normalized per regression target, as in (4). The widths of lines connecting data and parameters, and corresponding endpoints, are proportional to their respective values. In particular, the heights of the respective left end-points are proportional to the assigned sensitivity weights $q_{i_y}$ (4). Data types, e.g. $y_{5,1}, \ldots, y_{5,10}$, and parameters with their exponents, e.g. $\theta_1^1, \ldots, \theta_1^4$, are grouped by colors.

properties as well as employed ABC-SMC population sizes $N$ and total budgets of numbers of simulations are given in Table 1.

T1, T3, and T4 are problems M3, M4, and M5 from [11], respectively an ODE model of a conversion reaction, and, based on application examples in [10], g-and-k distribution samples, and a Markov jump process model of a Lotka-Volterra predator-prey process. T2 consists of two observables, thereof $y_1 \sim \mathcal{N}(\theta, 0.1^2)$ informative and $y_2 \sim \mathcal{N}(0, 1^2)$ uninformative, with wide prior $\theta \sim \mathcal{N}(0, 100^2)$, also from [10]. T5 and T6 are variations of T3 and T4 with higher-dimensional data, based on application examples in [15]. T5 employs 100 order statistics out of 10,000 samples from a g-and-k distribution, with $U[0, 10]$ priors on the four parameters $A$, $B$, $g$, $k$, considering ground truth values $(A, B, g, k) = (3, 1, 2, 0.5)$. T6 employs noise-free

**Table 1. Test model properties: Identifier, short description, number of parameters $n_\theta$ and data points $n_y$, population size $N$ and maximum number of model simulations after which an analysis was terminated.**

| ID | Description | $n_\theta$ | $n_y$ | $N$ | Max. sim. |
|----|-------------|------------|-------|-----|-----------|
| T1 | Conversion reaction ODE model | 2 | 10 | 1000 | 250000 |
| T2 | One informative and one uninformative variable | 1 | 2 | 1000 | 25000 |
| T3 | g-and-k distribution order statistics, small | 4 | 7 | 1000 | 250000 |
| T4 | Lotka-Volterra Markov jump process model, small | 3 | 32 | 500 | 125000 |
| T5 | g-and-k distribution order statistics, large | 4 | 100 | 1000 | 250000 |
| T6 | Lotka-Volterra Markov jump process model, large | 3 | 200 | 500 | 125000 |

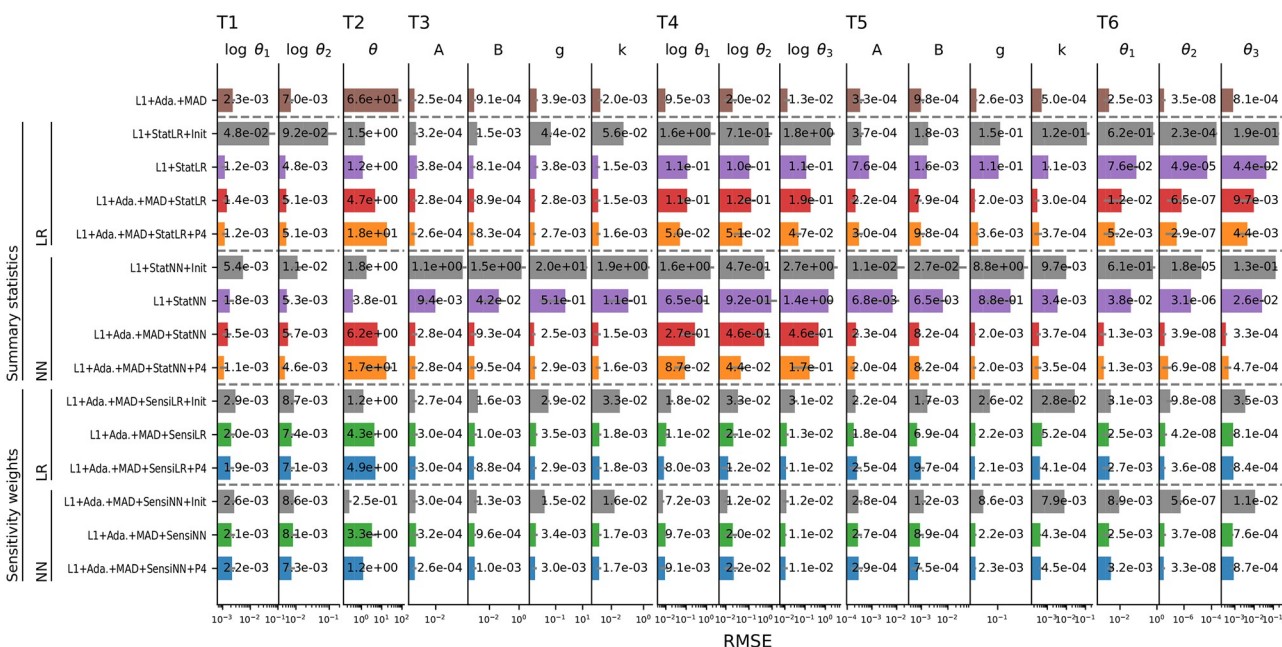

**Fig 4. Median RMSE (smaller is better) for the parameters of models T1–6 (columns) obtained for 15 inference methods (rows), using an L1 distance, either uniformly weighted if unspecified, or with adaptive MAD scale normalization ("Ada.+MAD").** As regression models we considered linear regression ("LR") and neural networks ("NN"), both to define summary statistics ("Stat") and sensitivity weights ("Sensi"). Some inference settings further used parameter transformations $\lambda(\theta) = (\theta^1, \ldots, \theta^4)$ as regression targets ("P4"). In some settings, the regression model was trained before the initial generation ("Init"), or after 40% of the simulation budget if unspecified. The first row contains solely scale-normalized L1 +Ada.+MAD as a reference, followed by two blocks of four rows using summary statistics, using firstly LR and secondly NN, and then by two blocks of three using sensitivity weights, using firstly LR and secondly NN. Reported values are medians over 10 replicates, with horizontal grey error lines indicating MAD.

observations of predators and prey at 200 evenly-spaced time-points over the interval [0, 20], estimating the three reaction rate coefficients on linear scale, considering tight independent priors $\theta_1 \sim U[0, 2]$, $\theta_2 \sim U[0, 0.1]$, $\theta_3 \sim U[0, 1]$, and ground truth values $(\theta_1, \theta_2, \theta_3) = (0.5, 0.0025, 0.3)$. We ran 10 repetitions of different inference scenarios on problems T1–6 on different data sets. To measure fit quality, we reported root mean square errors (RMSE) of the weighted posterior samples from the last ABC-SMC generation, with respect to ground truth parameters (note all problem considered here are uni-modal). The results are visualized in Fig 4.

**Delay of regression model training advantageous on complex models.** For the considered LR and NN models, regression model training on prior samples ("Init") gave for most problems substantially worse results than when trained after 40% of the simulation budget. One reason may be that only $N$ prior samples were used for training, compared to potentially more samples, including rejected ones, in later generations. However, also when using only $N$ training samples in the later-trained approach (not shown here), results were better than based on the prior. Thus, an explanation is that after multiple generations the bulk of samples is restricted to a high-density region, in which a simpler model is sufficiently accurate. This justifies empirically the approach by [15] of using a pilot run to constrain parameters. [16], who base their regression model on the prior, use firstly more complex NN models, and secondly up to 1$e$6 training samples, far more than entire analyses here. An exception was T2, where sometimes initial training improved performance. This can be explained by the global linear parameters-data mapping, such that accurate regression models can be easily learned and thereafter be beneficial.

**Scale normalization improves performance for regression-based summary statistics.** As the comparison of L1+Stat{LR/NN} and L1+Ada.+MAD+Stat{LR/NN} shows, the use of scale normalization improved performance for many problems, particularly for T5+6, while it was roughly similar for T1. An exception was again T2, where in fact a uniformly weighted L1 distance would be preferable over L1+Ada.+MAD at least in the first generations, as the uninformative observable happens to vary less there. It should further be remarked that overall, L1+Ada.+MAD performed quite robustly across all problems, including high-dimensional problems T5+6, and was consistently beaten by the methods accounting for informativeness only on T2. This again highlights the importance of scale normalization, and suggests that on these problems there were no overall uninformative data points (except T2) confounding the analysis.

**Sensitivity-weighted distances perform highly robustly.** The approaches L1+Ada.+MAD+Sensi{LR/NN}(+P4) using regression models to define sensitivity weights performed reliably, with RMSE values generally not far higher, but in some cases consistently lower, than those obtained by L1+Ada.+MAD. This indicates that, while the sensitivity weighting could in those cases not improve performance, as sole scale normalization was efficient already, the approach is highly robust. In some cases, specifically T2, which had one clearly uninformative statistic, and arguably T5, which is a high-dimensional collection of order statistics, did the sensitivity weighting improve performance. In other cases, specifically T1, T3, T4, and T6, RMSE values for some parameters decreased, but slightly increased for others, indicating that the weighting scheme re-prioritized data points, while no overall uninformative ones could be disregarded.

**Regression-based summary statistics can be superior but also less robust.** In various cases, e.g. when trained in the initial generation, and consistently for T4, as well as using LR on T6, summary statistics were inferior to both L1+Ada.+MAD and sensitivity weights. Arguably, in those cases the regression model was not accurate enough to allow using its outputs as low-dimensional summaries. However, in some cases, specifically for T1, and two parameters of T6 using NN, RMSE values obtained using summary statistics were smaller than with both L1+Ada.+MAD and sensitivity weights. This again indicates that if the lower-dimensional summary statistics representation is accurate and informative of the parameters, then its use can be beneficial and superior to mere re-weighting.

**No clear preference for regression model or target augmentation.** For both regression-based summary statistics and sensitivity weights, we found overall no clear preference for LR or NN, with LR more robust in many cases, but NN clearly preferable in some. Further, the use of augmented parameters as regression targets did not substantially worsen, but also not notably improve performance for any test problem, however performed inferior e.g. on T2, which has a clear linear mapping, such that the consideration of higher-order moments may have complicated the inference. This indicates that using augmented parameters as regression targets is robust, but if further information is available, a restriction to e.g. first or second order may be beneficial.

## 3.4 Performance on application example

Next, we considered an agent-based model of tumor spheroid growth (model M6 from [11]), considering both outlier-free and outlier-corrupted data. We employed the same simulated data as in [11], a population size of $N = 500$, and a computational budget of 150,000 simulations per analysis. Given its computational complexity with ABC-SMC wall times on the order of several hours even with parallelization on hundreds of CPU cores, we had to restrict our analysis of this problem to only a few selected approaches: Besides the reference L1+Ada.

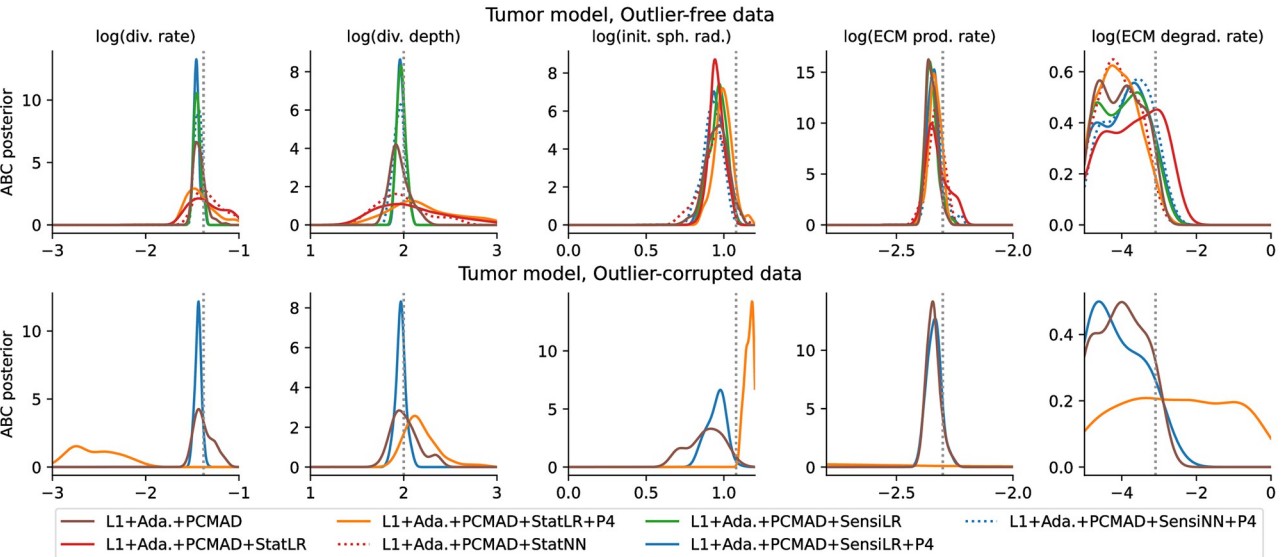

**Fig 5. Posterior marginals for 5 out of the 7 model parameters of the tumor problem with interesting behavior.** Without (top) and with (bottom) outliers. Ground truth parameters are indicated by vertical grey dotted lines. Plot boundaries are the employed uniform prior ranges, except the ECM production rate is zoomed in for visibility.

+PCMAD, we employed, given the robust performance of LR before, L1+Ada.+PCMAD+ SensiLR(+P4) using sensitivity weights, L1+Ada.+PCMAD+StatLR(+P4) using summary statistics, both with and without augmented regression targets $\lambda(\theta) = (\theta^1, \ldots, \theta^4)$, further L1 +Ada.+PCMAD+StatNN and L1+Ada.+PCMAD+SensiNN+P4 using NN. Here, to facilitate outlier detection, we used PCMAD instead of MAD.

**Sensitivity weights identify uninformative model outputs.** Using regression models to define sensitivity weights improved performance on the tumor model with outlier-free data over L1+Ada.+PCMAD, giving lower variances for the division rate and depth parameters, with otherwise similar results (Fig 5 top), and accepted simulations closely matching the observed data (Fig 6 top, simulations). No differences could be observed between using only the parameters, or also higher-order moments, as regression targets.

On this problem, regression-based summary statistics performed substantially worse, which may indicate that the employed regression models did not provide a sufficiently informative low-dimensional representation (L1+Ada.+PCMAD+StatLR(+P4), Fig 5 top), simulations did visibly not match the observed data (Fig 6 top, 1st row).

The overall structure of sensitivity weights assigned via LR with and without parameter augmentation, as well as NN, was roughly consistent across multiple runs (Fig 6 top, 3rd row). Low weights were assigned to the fraction of proliferating cells at large distances to the rim, indicating these to be uninformative, and counteracting the large weights resulting from scale normalization (Fig 6 top, 2nd row). While the sensitivity weights exhibit some variability between adjacent points and across runs, consistent and reasonable overall patterns can be observed.

**Robust on outlier-corrupted data.** Using sensitivity weights improved performance also on outlier-corrupted data (Fig 5 bottom). Given its previously good performance, here we only considered L1+Ada.+PCMAD+SensiLR+P4. Accepted simulations in the final generation matched the observed data more closely than for L1+Ada.+PCMAD (Fig 6 bottom, 1st row). The PCMAD scheme assigned low weights to outliers, independent of the regression-based

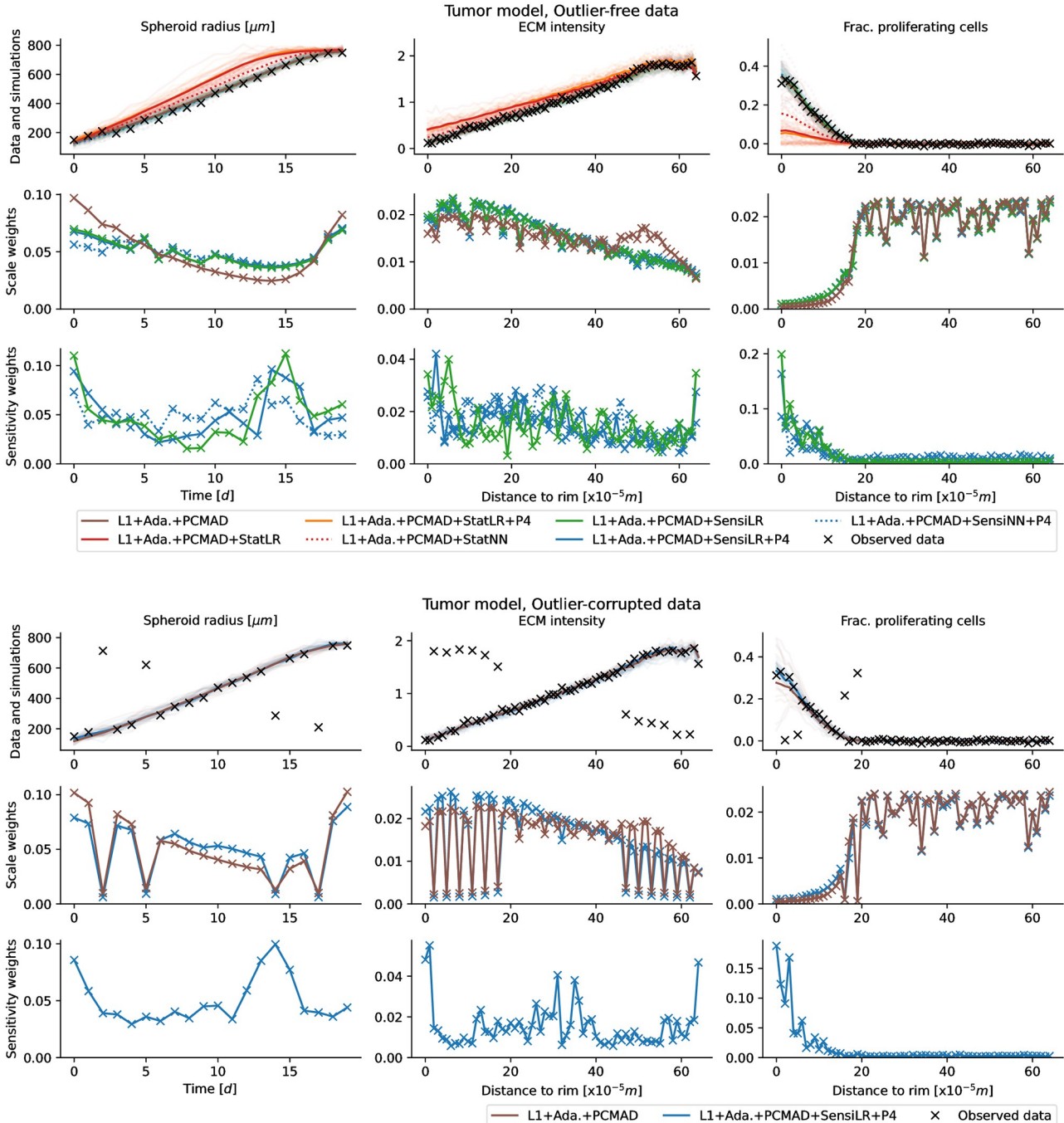

**Fig 6. Fits, scale and sensitivity weights for the tumor problem on outlier-free (top) and outlier-corrupted (bottom) data.** The respective upper rows show the observed data (black), and, for each approach, 20 accepted simulated data sets (light lines) as well as the sample means (darker lines) from the last ABC-SMC generation. The respective middle rows show the scale weights assigned to each data point in the last generation, normalized to unit sum, and the bottom rows the sensitivity weights, respectively only for distances employing such weights, and operating on the full data.

sensitivity weights (Fig 6 bottom, 2nd and 3rd row). Thus, the combination of both methods allowed to simultaneously account for outliers and informativeness. For an in-depth study of adaptive scale normalization in the presence of data outliers, see [11].

## 4 Discussion

In this work, we discussed problems arising in ABC (1) from partly uninformative data for scale-normalized distances, (2) from heterogeneous parameter scales for regression-based summary statistics, and (3) from parameter non-identifiability for regression model adequacy. To tackle these problems, we presented multiple solutions: First, we suggested employing adaptive scale-normalizing distances on top of regression-based summary statistics, to homogenize the impact of parameters. Second, as an alternative to the first solution, we introduced novel sensitivity weights derived from regression models, measuring the informativeness of data on parameters. Third, we introduced augmented regression targets to overcome parameter non-identifiability.

We showed substantial improvements of the novel methods over established approaches on a simple demonstration problem. For the sensitivity-weighted distances, we showed robust performance on various further test problems, in particular on a complex systems biological application problem. Yet, there are numerous ways in which the presented methods can be improved:

While simple linear models often sufficed, especially when trained on a high-density region, in some cases more complex models were superior, indicating the presence of non-linear data-parameter relationships. A systematic investigation of alternative and more complex regression model types, e.g. neural networks tailored to the respective data types, as well as model selection among competing regression models, would be useful. To select among competing model candidates, an efficient yet robust model selection scheme with out-of-sample evaluation, e.g. based on cross validation, might be useful. The employed regression model should reflect and exploit any symmetries or structures in the data. For example, for image and sequence data convolutional and recurrent networks would be prime candidates, for i.i.d. data permutation invariant networks [35]. More complex regression models and robust estimators require larger training sets (which could be mitigated by transfer learning). While increasing the training set is straightforward, as it only requires continued sampling from the forward model, there is a cost trade-off of the actual ABC inference and regression model training, which remains to be investigated.

While in many cases delaying regression model training to later generations and a high-density region was advantageous, for simple models we observed benefits of early regression. Besides training prior to the first generation, we only considered training the regression model after 40% of the simulation budget, based on the heuristic that the ABC-SMC algorithm should have had time to converge to a higher posterior density region, while still leaving a substantial simulation budget to leverage the learned mapping. However, in practice good training times will clearly be problem-specific. Criteria on if and when to train or update regression models, also repeatedly, would be of interest.

While augmenting regression targets was essential to render inverse regression model based approaches applicable to non-identifiable forward models, one could further study the ramifications of the increased summary statistics dimension, as well as methods to automatically adjust the set of applied transformations. In particular, for multivariate problems it may be necessary to also consider combinations of parameters to render the inverse problem identifiable. While we showed that our approach also performed robustly in the presence of outliers

for a given model, its applicability in general under model misspecification could be studied further [36, 37].

It should be noted that in calculating sensitivity weights we make the assumption that the employed regression model is (sub-)differentiable, which clearly holds for the regression models employed in this work (see also S1 File, Section 3). Further, we implicitly assume that the employed smooth regression model provides an accurate approximation of the underlying expectations it aims to describe. In practice, it should be carefully evaluated whether such an approximation appears valid for the given forward model. In the first place, we use the derived sensitivity matrix as a "heuristic" to quantify informativeness for weighting. However, it may fail in this function as a heuristic if the underlying quantities are inadequately captured. In particular, it provides a biased approximation of the derivative it intends to approximate when in the underlying expectations integration and differentiation cannot be exchanged in finite samples. Studying the consequences of such conceptual and practical discrepancies as well as investigating alternative measures of sensitivity could improve robustness and applicability of the method.

This work may be regarded as an extension of the approaches of [10, 11] as well as [15]. An alternative weighting scheme is presented by [38], who maximize a distance between samples from the prior and the posterior approximation. While using a different notion of informativeness and a specific underlying sampler, a comparison in terms of efficiency, robustness to outliers, and information gain would be of interest. A core problem with model selection in ABC is that insufficient summary statistics can lead to inconsistent Bayes factors [39]. Thus, approaches such as our adaptive weights accounting for scale and informativeness, or also [38, 40], hold the potential to facilitate unbiased ABC model selection even on larger data sets.

We have shown here that adaptive distance metrics accounting for both scale and informativeness facilitate an efficient analysis. We envisage that they may prove valuable also in combination with many orthogonal methods in an ABC-SMC framework. Promising appear for example a combination with optimal thresholding schemes [26], as well as a combination with multifidelity approaches using surrogates to even further reduce the number of evaluations of expensive simulators [41].

All methods presented in this work have been implemented in the Python package pyABC, facilitating their straightforward application. We anticipate that such approaches, which automatically normalize and extract or weight features of interest without extensive manual tuning, will substantially improve performance of ABC methods on a wide range of applications problems.

## Supporting information

**S1 File. Contains all the supporting text and figures.**
(PDF)

## Acknowledgments

We thank Dennis Prangle and Marc Vaisband for fruitful discussions, Emad Alamoudi for help with the HPC setup, and acknowledge the Gauss Centre for Supercomputing for providing computing time on the GCS Supercomputer JUWELS [42] at Jülich Supercomputing Centre.

## Author Contributions

**Conceptualization:** Yannik Schälte, Jan Hasenauer.

**Formal analysis:** Yannik Schälte.

**Funding acquisition:** Jan Hasenauer.

**Investigation:** Yannik Schälte.

**Methodology:** Yannik Schälte.

**Project administration:** Jan Hasenauer.

**Software:** Yannik Schälte.

**Supervision:** Jan Hasenauer.

**Validation:** Yannik Schälte.

**Visualization:** Yannik Schälte.

**Writing – original draft:** Yannik Schälte.

**Writing – review & editing:** Yannik Schälte, Jan Hasenauer.

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
