## [Decision Letter · Decision Letter 0]

21 Nov 2022

PONE-D-22-20368Informative and adaptive distances and summary statistics in sequential approximate Bayesian computationPLOS ONE

Dear Dr. Schälte,

Thank you for submitting your manuscript to PLOS ONE. After careful consideration, we feel that it has merit but does not fully meet PLOS ONE’s publication criteria as it currently stands. Therefore, we invite you to submit a revised version of the manuscript that addresses the points raised during the review process.

ACADEMIC EDITOR: I see that this work has a potential for publication in PLOS ONE. But I see that several issues posted by all reviewers must be clearly addressed before reconsideration.

We look forward to receiving your revised manuscript.

Kind regards,

Nattapol Aunsri, Ph.D.

Academic Editor

PLOS ONE

Journal Requirements:

2. Please note that PLOS ONE has specific guidelines on code sharing for submissions in which author-generated code underpins the findings in the manuscript. In these cases, all author-generated code must be made available without restrictions upon publication of the work. Please review our guidelines at https://journals.plos.org/plosone/s/materials-and-software-sharing#loc-sharing-code and ensure that your code is shared in a way that follows best practice and facilitates reproducibility and reuse. New software must comply with the Open Source Definition.

Reviewers' comments:

Reviewer's Responses to Questions

**Comments to the Author**

1. Is the manuscript technically sound, and do the data support the conclusions?

Reviewer #1: Yes

Reviewer #2: Yes

Reviewer #3: Yes

2. Has the statistical analysis been performed appropriately and rigorously? 

Reviewer #1: Yes

Reviewer #2: Yes

Reviewer #3: Yes

3. Have the authors made all data underlying the findings in their manuscript fully available?

Reviewer #1: Yes

Reviewer #2: Yes

Reviewer #3: Yes

4. Is the manuscript presented in an intelligible fashion and written in standard English?

Reviewer #1: Yes

Reviewer #2: Yes

Reviewer #3: Yes

5. Review Comments to the Author

Reviewer #1: Please see the attached comments.

Reviewer #2: Review report of ``Informative and adaptive distances and summary statistics in sequential approximate Bayesian computation''

by Yannik Schalte and Jan Hasenauer

Summary: In this paper the authors present a method to adaptively reweight terms in the distance metric used in approximate Bayesian computation.

These reweighting approach exploits sequential Monte Carlo (SMC) for ABC byt using early iterations to learn the inverse mapping from the

data or summary statistics to parameters. For the training steps they consider both statistical (i.e., linera regression) and machine learing

(i.e., neural networks). The key elements of there method is demonstrated on a specially constructed toy problem and then tested on a raneg of

common benchmark models for ABC methods. Consistently their new approach is shown to have greater statistical efficiency and, more importantly,

better robustness properties in the situation where data is corrupted by outliers.

I thoroughly enjoyed reading this paper as it addresses a very important problem for approximate Bayesian computation, that is the choice of distance

metric and summary statistics. It was also refreshing to the see the problem addressed for the cases when the inverse mapping does not exist due to

idendifiability concerns. While the statistically efficiency improvements are not always substantial, the robustness properties of this approach alone

with its applicability to non-identifiable models make this a very useful piece of work. I therefore recommend it for publication pending some minor

revisions based on my comments below.

1. Regarding the main contribution (Algorithm 2), I have a few minor comments:

1.1 What tuning is involved in selecting the value for t_train? In the paper it seems that the only

choices considered are t_train = 1 and t_train such that 40% of computational budget is used. How was

the 40% budget chosen and what other considerations are important for choosing t_train?

1.2 Throughout only L1 is used based on it's robustness properties. It would be interesting to see an examples of

this adaptive scheme under L2 or another Lp in the presence of outliers. Given the discussed robustness

properties of the adaptive reweighting, presumably this would better highlight this property compared to other

approaches with L2 or other Lp distances.

1.3 I believe there is a typo in the second line of Algorithm 2 (for t = t_train, ..., n_t do), as the conditions

in the subsquent if conditions (i.e., "if t < t_train then" and "if t+1 == t_trian") will alwasy be false.

2. In section 2.2.3 when using multiple parameter moments to account for identifiability I am not completely clear on

the implementation. Are you a) fitting k summaries independently (i.e., s_i from y ~ theta^i) or b) using all the moments

to obtain a single summary (i.e., s from y ~ theta + theta^1 + ... + theta^k).

3. For the toy example, it is not completely clear how the synthetic dataset is constructed. Is each observsation a) a

5-d vector (with each component from the respective distibution), or b) a scalar randomly chosen from one of the

components. If the answer is b) is each component equally likely, or are some rare events?

4. The following questions relate to the results:

4.1 In the results table in Figure 4, the vanilla L1+Ada.+MAD works surprisingly well in some cases. Can you provide some insight

into this? I also presume that these results are for equivalent computational budgets.

4.2 For the tumor problem under the corruption of outliers why are only L1+Ada.+PCMAD+StatLR+P4 and L1+Ada.+PCMAD+SensiLR+P4 applied?

5. There are a few discussion points that could be worth considering:

5.1. How do you deal with the case when rare "outlier" events are part of the data generating process.

5.2. beyond corruption of outliers, there are other kinds of miss-specification that are structural (e.g., using an SIR model when

the real process is SEIR). Could your approach be used to identify this?

5.3 I believe this adaptive scheme could be particularly valuable for multifidelity method which are methods that exploit approximate models

and appropriately correct for bias. Could your method be applied to automatically construct

summaries/distances for each model fidelity, such that the ROC properties of the approximations are improved (and therefore less need to

simulated the exact model). This could be particularly useful within Adaptive mutlifidelity schemes such as

https://doi.org/10.1137/20M1316160

https://arxiv.org/abs/2112.11971

https://doi.org/10.1016/j.jcp.2022.111543

6. Some minor typographical things throughout that I found, suggest a careful proof-read:

6.1 Line 30 "allowing to understand"

6.2 Line 33 "allows doing so"

6.3 Line 37 "In a nutshell" is a bit informal suggest "Put briefly"

6.4 Line 39 "This way," -> "In this way,"

6.5 Line 44 "demonstrate" -> "demonstrates"

6.6 Lines 54-55 sentence is a bit awkward, suggest rephrasing.

6.7 throughout section 2: define all maths symbols e.g., n_y n_s and n_theta are never defined

6.8 Line 105 "Monte-Carlo" -> "Monte Carlo"

6.9 Line 214 define PCMAD

Reviewer #3: Using NN to improve summary statistics of improve ABC is a nice idea. I posed three questions in my review of the manuscirpt concerning non-identifiability, curse of dimensionality and conditions for model selection.

6. PLOS authors have the option to publish the peer review history of their article (what does this mean?). If published, this will include your full peer review and any attached files.

Reviewer #1: No

Reviewer #2: No

Reviewer #3: **Yes: **Marcos A. Capistran

---

## [Author Response · Author response to Decision Letter 0]

27 Jan 2023

Dear reviewers,

We thank you very much for your in-depth assessment of our manuscript. We have addressed all comments and incorporated changes which definitely improved the quality of our manuscript. Please find attached our one-by-one response, as well as a revised version of the manuscript.

Kind regards,

Yannik Schaelte

---

## [Decision Letter · Decision Letter 1]

28 Feb 2023

PONE-D-22-20368R1Informative and adaptive distances and summary statistics in sequential approximate Bayesian computationPLOS ONE

Dear Dr. Schälte,

Thank you for submitting your manuscript to PLOS ONE. After careful consideration, we feel that it has merit but does not fully meet PLOS ONE’s publication criteria as it currently stands. Therefore, we invite you to submit a revised version of the manuscript that addresses the points raised during the review process.

ACADEMIC EDITOR: Please consider all reviewer's comments carefully before submitting your revised manuscript.

We look forward to receiving your revised manuscript.

Kind regards,

Nattapol Aunsri, Ph.D.

Academic Editor

PLOS ONE

Journal Requirements:

Reviewers' comments:

Reviewer's Responses to Questions

**Comments to the Author**

1. If the authors have adequately addressed your comments raised in a previous round of review and you feel that this manuscript is now acceptable for publication, you may indicate that here to bypass the “Comments to the Author” section, enter your conflict of interest statement in the “Confidential to Editor” section, and submit your "Accept" recommendation.

Reviewer #1: All comments have been addressed

Reviewer #2: All comments have been addressed

Reviewer #3: All comments have been addressed

2. Is the manuscript technically sound, and do the data support the conclusions?

Reviewer #1: Yes

Reviewer #2: Yes

Reviewer #3: Yes

3. Has the statistical analysis been performed appropriately and rigorously? 

Reviewer #1: Yes

Reviewer #2: Yes

Reviewer #3: Yes

4. Have the authors made all data underlying the findings in their manuscript fully available?

Reviewer #1: Yes

Reviewer #2: Yes

Reviewer #3: Yes

5. Is the manuscript presented in an intelligible fashion and written in standard English?

Reviewer #1: Yes

Reviewer #2: Yes

Reviewer #3: Yes

6. Review Comments to the Author

Reviewer #1: Please see the attached comments.

Reviewer #2: I am very pleased with the responses by the authors. All my questions and comments have been addressed.

I believe this will be a very useful paper to the systems biology community, and statistical computing more broadly.

Reviewer #3: The authors have addressed all my concerns. I appreciate their careful and insightful replies. In particular, the one about model selection.

7. PLOS authors have the option to publish the peer review history of their article (what does this mean?). If published, this will include your full peer review and any attached files.

Reviewer #1: No

Reviewer #2: No

Reviewer #3: **Yes: **Marcos A. Capistran

---

## [Author Response · Author response to Decision Letter 1]

17 Apr 2023

Dear reviewers,

Thank you very much for your previous feedback. We hope to have addressed the remaining comments of Reviewer 1.

Best,

Yannik Schaelte

---

## [Decision Letter · Decision Letter 2]

3 May 2023

Informative and adaptive distances and summary statistics in sequential approximate Bayesian computation

PONE-D-22-20368R2

Dear Dr. Schälte,

We’re pleased to inform you that your manuscript has been judged scientifically suitable for publication and will be formally accepted for publication once it meets all outstanding technical requirements.

Kind regards,

Nattapol Aunsri, Ph.D.

Academic Editor

PLOS ONE

Additional Editor Comments (optional):

Reviewers' comments:

Reviewer's Responses to Questions

**Comments to the Author**

1. If the authors have adequately addressed your comments raised in a previous round of review and you feel that this manuscript is now acceptable for publication, you may indicate that here to bypass the “Comments to the Author” section, enter your conflict of interest statement in the “Confidential to Editor” section, and submit your "Accept" recommendation.

Reviewer #1: All comments have been addressed

2. Is the manuscript technically sound, and do the data support the conclusions?

Reviewer #1: Yes

3. Has the statistical analysis been performed appropriately and rigorously? 

Reviewer #1: Yes

4. Have the authors made all data underlying the findings in their manuscript fully available?

Reviewer #1: Yes

5. Is the manuscript presented in an intelligible fashion and written in standard English?

Reviewer #1: Yes

6. Review Comments to the Author

Reviewer #1: The authors have adequately addressed my remaining concerns. I have no remaining comments.

7. PLOS authors have the option to publish the peer review history of their article (what does this mean?). If published, this will include your full peer review and any attached files.

Reviewer #1: No

---

## [Editor Report · Acceptance letter]

11 May 2023

PONE-D-22-20368R2 

Informative and adaptive distances and summary statistics in sequential approximate Bayesian computation 

Dear Dr. Hasenauer:

I'm pleased to inform you that your manuscript has been deemed suitable for publication in PLOS ONE. Congratulations! Your manuscript is now with our production department. 

Kind regards, 

on behalf of

Dr. Nattapol Aunsri 

Academic Editor

PLOS ONE